# Adenovirus-Based Gene Therapy for Bone Regeneration: A Comparative Analysis of In Vivo and Ex Vivo *BMP2* Gene Delivery

**DOI:** 10.3390/cells12131762

**Published:** 2023-07-01

**Authors:** Tatiana Borisovna Bukharova, Irina Alekseevna Nedorubova, Viktoria Olegovna Mokrousova, Anastasiia Yurevna Meglei, Viktoriia Pavlovna Basina, Andrey Anatolevich Nedorubov, Andrey Vyacheslavovich Vasilyev, Timofei Evgenevich Grigoriev, Yuriy Dmitrievich Zagoskin, Sergei Nicolaevich Chvalun, Sergey Ivanovich Kutsev, Dmitry Vadimovich Goldshtein

**Affiliations:** 1Research Centre for Medical Genetics, 115478 Moscow, Russia; nedorubova.ia@gmail.com (I.A.N.); victoria-mok@yandex.ru (V.O.M.); an.megley95@yandex.ru (A.Y.M.); vika.basina12@gmail.com (V.P.B.); vav-stom@yandex.ru (A.V.V.); kutsev@mail.ru (S.I.K.); dvgoldshtein@gmail.com (D.V.G.); 2Institute of Translational Medicine and Biotechnology and E.V. Borovsky Institute of Dentistry, I.M. Sechenov First Moscow State Medical University of the Ministry of Health of the Russian Federation (Sechenov University), 119991 Moscow, Russia; nedorubov.ras@gmail.com; 3NRC “Kurchatov Institute”, 123182 Moscow, Russia; timgrigo@yandex.ru (T.E.G.); zagos@inbox.ru (Y.D.Z.); chvalun_sn@nrcki.ru (S.N.C.)

**Keywords:** gene therapy, adenoviral transduction, mesenchymal stem cells, polylactide granules, platelet rich plasma, bone regeneration

## Abstract

Adenovirus-mediated gene therapy is a promising tool in bone regenerative medicine. In this work, gene-activated matrices (GAMs) composed of (1) polylactide granules (PLA), which serve as a depot for genetic constructs or matrices for cell attachment, (2) a PRP-based fibrin clot, which is a source of growth factors and a binding gel, and (3) a *BMP2* gene providing osteoinductive properties were studied. The study aims to compare the effectiveness of in vivo and ex vivo gene therapy based on adenoviral constructs with the *BMP2* gene, PLA particles, and a fibrin clot for bone defect healing. GAMs with Ad-BMP2 and MSC(Ad-BMP2) show osteoinductive properties both in vitro and in vivo. However, MSCs incubated with GAMs containing transduced cells showed a more significant increase in osteopontin gene expression, protein production, Alpl activity, and matrix mineralization. Implantation of the studied matrices into critical-size calvarial defects after 56 days promotes the formation of young bone. The efficiency of neoosteogenesis and the volume fraction of newly formed bone tissue are higher with PLA/PRP-MSC(Ad-BMP2) implantation (33%) than PLA/PRP-Ad-BMP2 (28%). Thus, ex vivo adenoviral gene therapy with the *BMP2* gene has proven to be a more effective approach than the in vivo delivery of gene constructs for bone regeneration.

## 1. Introduction

The active development of genetic technologies for regenerative medicine [1] and the appearance of examples of the successful use of various genetic constructs for therapeutic purposes highlight the need to substantiate the effectiveness and safety of various gene therapy strategies. The most promising strategies for the delivery of genetic information are viral methods based on the natural ability of viral particles to penetrate cell membranes and ensure efficient expression of target genes in cells [2,3,4]. Recombinant adenoviral vectors are receiving increasing attention due to their lack of replication function and inability to integrate into the host cell genome, ensuring a sufficiently high level of safety for their use [5]. Additionally, adenoviruses can transduce different types of cells, both dividing and non-dividing, and are characterized by a high packing capacity, which is their advantage over other commonly used vectors [6]. The active use of adenoviral vector-based vaccines against coronavirus has shown their ability to ensure the production of target proteins at a high level in a large population of patients during a pandemic, with low rates of complications from immune therapy [7].

When developing gene therapy strategies for bone regeneration, the use of bone morphogenetic protein 2 (*BMP2*) as a therapeutic gene is the most promising [8]. The product of this gene is one of the most effective osteoinductors [9]. BMP-2 interacts with a receptor on the membrane of osteogenic precursors, triggering signaling pathways that regulate the development of osteogenesis. The treatment of bone defects by gene therapy requires the use of carrier matrices to ensure the targeted delivery of genetic constructs or transduced cells to the damaged area in order to achieve a high therapeutic concentration of osteoinductors and minimize the risk of heterotopic osteogenesis outside the regeneration site [10]. Highly porous polylactide particles (PLA), proven to be a biocompatible material with a high capacity for filling with gene constructs and proteins [11,12,13,14], can serve as a promising material for depositing genetic vectors. The efficiency of bone regeneration largely depends on the rate of vascularization [15,16]. The natural source of angiogenic factors, as well as a number of other growth factors and cytokines, is platelet-rich plasma (PRP), that can be easily obtained from the patient, does not require purification from antigens and foreign components and is completely compatible with the patient’s organism [17]. Polymerization of fibrinogen contained in PRP makes it possible to obtain a fibrin clot, inside which matrices components can be enclosed. The use of PRP-based fibrin clots in bone therapy has proven to be useful [18], but as carrier matrices for viral vectors, it is used in single works and requires additional research due to the high probability of implementation in clinical practice.

The effectiveness of bone defect treatment using viral constructs largely depends on whether the cells are transduced in vivo or ex vivo. During in vivo gene therapy, viral vectors are introduced into the body, which can penetrate resident cells, that then synthesize target inducer proteins. This leads to the induction of osteogenic differentiation of the patient’s own osteoprogenitor cells of mesenchymal origin. In ex vivo gene therapy, multipotent mesenchymal stem cells (MSCs) are typically used, which are genetically modified to synthesize target protein inducers. These cells are a source of osteogenic factors that can affect both resident and transplanted cells [2,3,8]. The development of two therapeutic approaches and a comparison of their effectiveness in vivo and in vitro within the framework of one experimental study will help to understand the mechanisms of osteogenesis and reveal the advantages and disadvantages of different methods of adenovirus therapy for bone defects based on the *BMP2* gene.

The aim of this work was to compare the effectiveness of in vivo and ex vivo gene therapy based on adenoviral constructs with the *BMP2* gene, PLA particles, and a fibrin clot obtained from PRP for bone defect healing.

## 2. Materials and Methods

### 2.1. Matrices

Polylactide granules (PLA) were obtained by freeze-drying poly-L-lactide with a molecular weight of 200 kDa (4032D, NatureWorks, Plymouth, MN, USA). PLA granules with a diameter of 50–200 μm had a porosity of 98% and a pore diameter of 2–10 µm [19]. Before testing, PLA granules were subjected to sterilization with 70% ethanol for 30 min, followed by a 1 h wash in physiological saline (PanEco, Moscow, Russia).

Platelet-rich plasma (PRP) was obtained according to the previously described method [11]. To obtain PLA/PRP matrices, PLA were mixed with PRP in a volume ratio of 1:1. Then, thrombin solution (PZ Cormay, Warsaw, Poland) in 10% calcium chloride solution was added dropwise for polymerization.

To obtain gene-activated matrices, 1000 TCID50/mL adenovirus constructs with the *BMP2* gene (Ad-BMP2) or *GFP* gene (Ad-GFP) were incubated with PLA granules for 1 h at 37 °C, mixed with PRP and polymerized with thrombin solution. To obtain matrices impregnated with transduced cells, PLA granules were incubated with 10 × 10^6^/mL MSC(Ad-GFP) or MSC(Ad-BMP2) for 5 min at 37 °C, mixed with PRP and polymerized with thrombin solution.

### 2.2. Cell Cultures

MCSs derived from rat adipose tissue using a previously developed procedure [20] were cultured in a growth medium of DMEM/F12 (PanEco, Moscow, Russia) containing 10% fetal bovine serum (FBS; PAA Laboratories, Etobicoke, ON, Canada), 0.584 mg/mL L-glutamine (PanEco, Moscow, Russia), 5000 U/mL penicillin (PanEco, Moscow, Russia) and 5000 mg/mL streptomycin (PanEco, Moscow, Russia) under standard culture conditions at 37 °C and 5% CO_2_. The culture medium was changed every three days.

MSCs were transduced with Ad-GFP or Ad-BMP2 in DMEM/F12 medium with antibiotics and 2% FBS for 6, 16 or 24 h with a viral load of 80, 160 or 320 TCID50/mL, depending on the experimental conditions.

Before MSCs impregnation into the matrices, the cells were transduced using 320 TCID50/mL adenovirus constructs for 24 h. Then, the cells were detached from the surface using a Versene solution (PanEco, Moscow, Russia) with the addition of 0.25% trypsin (PanEco, Moscow, Russia), centrifuged at 1200 rpm and impregnated into the matrices.

For osteogenic differentiation, MSCs were cultivated in a medium supplemented with L-ascorbic acid (Sigma Aldrich, St. Louis, MO, USA) and 2.16 mg/mL glycerophosphate (Sigma Aldrich, St. Louis, MO, USA). Half of the medium was changed every three days during the experiment. MSCs cultured in an osteogenic medium served as a negative control.

### 2.3. MTT Test

Cells were incubated with 0.5 mg/mL MTT (3-(4,5-dimethylthiazol-2-yl)-2,5-diphenyltetrazolium bromide, PanEco, Moscow, Russia) for 2 h at 37 °C. After that, formazan crystals were extracted from the cells using DMSO (PanEco, Moscow, Russia), followed by shaking for 20 min. The optical density was measured on an xMark plate spectrophotometer (Bio-Rad, Hercules, CA, USA) at a wavelength of 570 nm, subtracting the background value at 620 nm. The results were compared to the control, which was taken as 100%.

### 2.4. Transduction Efficiency

Transduction efficiency Ad-GFP was assessed by the presence of the fluorescent GFP protein in cells using a fluorescence Axio Observer D1 microscope with an AxioCam HRc camera (Carl Zeiss Microscopy, GmbH, Oberkochen, Germany) and by flow cytometry on a CyFlow ML (Partec, Tsuchiura, Japan).

Transduction efficiency Ad-BMP2 was assessed by *BMP2* gene expression by RT-PCR and BMP-2 protein production by enzyme-linked immunosorbent assay (ELISA).

RT-PCR was carried out in a BioRad iQ cycler thermal cycler (BioRad, Hercules, CA, USA) using SYBR Green I intercalating dye (Eurogen, Moscow, Russia). Total RNA was extracted from the cells using the RNeasy Plus Mini kit (Qiagen, Hilden, Germany) and reverse transcribed into cDNA using the RevertAid Kit (Thermo Scientific, Bremen, Germany). The expression levels of the analyzed gene were normalized to the average values of the reference genes *Gapdh* and *Actβ*.

To evaluate the production of the BMP-2 protein by MSCs, the culture medium was collected every 3 days and stored at −80 °C. Then, all fractions were combined and concentrated using a 3 kDa Amicon spin column (Merck KGaA, Darmstadt, Germany). The protein was analyzed by ELISA using a Quantikine Elisa kit (R&D Systems, Minneapolis, MN, USA) following the manufacturer’s protocol. The measurements were performed on an xMark plate spectrophotometer.

### 2.5. Osteogenic Differentiation of MSCs

To determine the effectiveness of osteogenic differentiation of MSCs, the osteogenic markers’ gene expression (*Runx2*, *Spp1* and *Bglap*), alkaline phosphatase (Alpl) activity, osteopontin (Opn) protein production and extracellular matrix (ECM) mineralization of MSCs were studied.

Alpl activity was determined in cell lysates using the Quantitative Alkaline Phosphatase ES Characterization Kit (Merck KGaA, Darmstadt, Germany) according to the manufacturer’s instructions. The measurements were performed on an xMark plate spectrophotometer.

Opn protein production was analyzed by ELISA using an Elisa kit for osteopontin (Cloud-Clone Corp., Shanghai, China) according to the manufacturer’s protocol. The measurements were carried out on an xMark plate spectrophotometer.

ECM mineralization was detected by fixing MCSs with cooled 70% ethanol for 30 min at +4 °C and staining them with 2% aqueous solution of alizarin red (Sigma Aldrich, St. Louis, MO, USA) at pH 4.1 for 5 min. Unbound dye was removed by washing the cells twice with distilled water, and the images were obtained using light microscopy.

### 2.6. Matrices’ Cytocompatibility In Vitro

The matrices’ cytocompatibility in vitro was assessed using the MTT test and fluorescence microscopy. MCSs were seeded on the bottom of 24-well Transwell system plates (pore diameter 8 µm, SPL Lifesciences, Suwon, Republic of Korea) at a density of 25 × 10^3^ cells/mL, and the matrices were placed in the filters. Live cells were stained with 0.5 µM calcein AM (Biotium, Fremont, CA, USA) for 35 min at 37 °C, while dead cells were stained with 5 µg/mL DAPI (4,6-diamidino-2-phenylindole) for 10 min at 37 °C. Then, images were obtained using a fluorescence microscope.

To evaluate the adhesion of MCSs on the matrices, cells were seeded on the surface of the matrices and fixed in a 2.5% solution of glutaraldehyde (Panreac, Chicago, IL, USA) for 12 h at 4 °C after 1 and 7 d. Then, the samples were dehydrated in ethanol battery with increasing concentrations (50%, 75%, 80%, 90% and absolute ethanol) at 4 °C and analyzed via scanning electron microscopy (SEM) using a Phenom ProX microscope (Phenom, Rotterdam, The Netherlands) with an accelerating voltage of 15 kV.

### 2.7. In Vivo Studies

All in vivo experiments were approved by the local bioethical committee of Sechenov University (No. PRC-079 from 6 April 2021) in compliance with the Guide for the Care and Use of Laboratory Animals published by the US National Institutes of Health (NIH publication no. 85–23, revised 1996), the European Convention for the Protection of Vertebrate Animals used for Experimental and Other Scientific Purposes, and ISO 10993-2-2009.

The bone regeneration process in vivo was examined using a critical-sized calvarial defect model in Wistar rats weighing 250–300 g. Each experimental group consisted of six animals. The rats were anesthetized using intramuscular injection of 30 mg/kg Zoletil (Virbac, Carros, France) and 5 mg/kg Xylazine (InterchemieWerken “de Adelaar” BV, Netherlands). A cranial bone defect with a 7 mm diameter was created using a Creamer trepan (LZQ, Seoul, Republic of Korea) with sterile regular saline irrigation. The matrices were implanted into the bone defect, and after implantation, the periosteum and skin were sutured with Vicryl 5/0 (Ethicon, Raritan, NJ, USA). All rats were euthanized by CO_2_ inhalation 8 weeks after surgery, and the implantation sites were removed, fixed in 10% neutral formalin solution (Labiko, St. Petersburg, Russia) for at least 24 h and analyzed via histological examination and micro-CT.

### 2.8. Micro-CT

The new bone formation in the critical-size calvarial defect areas was assessed using a high-resolution micro-CT scanner (SkyScan 1276, Bruker, Kontich, Belgium) with an X-ray voltage of 60 kV. NRecon reconstruction software (Version 1.6.10.2, Bruker, Belgium) was used to generate a 3D reconstruction from a set of scanned images. The obtained images were analyzed using the Dragonfly software (Montreal, QC, Canada) and the new bone volume (Nb.V.%) was calculated.

### 2.9. Histological Assay

After micro-CT, the bone biopsy specimens were decalcified in EDTA for 3 weeks, dehydrated in a gradient of alcohols and xylene and embedded in paraffin. Sections with a thickness of 5–10 µm were prepared and stained with hematoxylin and eosin (H&E) and Masson’s trichrome staining (Biovitrum, Tomsk, Russia) according to the manufacturer’s protocol. The histological preparations were examined using an Axio Observer D1 microscope with an AxioCam HRc Axioimager M.1 camera. To assess the ratio of foreign body giant cells to PLA granule area, 20 fields of view were obtained from different areas of the defect. To assess the area of the newly formed bone (Nb.Ar) to defect area ratio, scans of the whole defect were obtained.

### 2.10. Statistical Analysis

The SigmaPlot 12.0 software (Systat Software Inc., Palo Alto, Santa Clara, CA, USA) was used for statistical analysis and graphing. All data are presented as µ ± SD. Intergroup differences were evaluated by one-way ANOVA using Tukey post hoc tests. Differences with *p* < 0.05 were considered statistically significant.

## 3. Results

### 3.1. Selection of Conditions for Efficient Adenoviral Transduction of MSCs

MSCs were incubated with adenoviral vectors at various concentrations and for various periods to determine the transduction conditions. Visually, the fluorescence of GFP-producing cells shows that an increase in the viral load and incubation time of MSCs with Ad-GFP promotes more efficient adenoviral cell transduction after 1 day (Figure 1a) and 3 days (Figure 1b). Incubation of cells with Ad-GFP at a dose of 80 TCID50/mL for 24 h resulted in the transduction of 9.7% of cells after 1 day (Figure 1c) and 34.1% of cells after 3 days (Figure 1d). When using 320 TCID50/mL Ad-GFP, transduction of 60.1% of cells was observed after 1 day (Figure 1c) and 84.4% after 3 days (Figure 1d).

As the viral load increases, cell death also increases (Figure 2a,b), and after 3 days, the number of viable cells was 86.4 ± 2.8% at a viral load of 320 TCID50/mL and 24 h of incubation (Figure 2b). A decrease in viability of more than 15% is significant, which led to the conclusion that it is not advisable to increase the viral load or incubation time any further.

Thus, the most optimal conditions for the adenoviral transduction of MSCs was incubation of cells with Ad-GFP at a concentration of 320 TCID50/mL for 24 h.

The conditions chosen for the transduction of MSCs with Ad-GFP model vectors were applied to experiments with adenoviruses carrying the target *BMP2* gene (Ad-BMP2). It was shown that the transduction of MSCs with 320 TCID50/mL Ad-BMP2 for 24 h had a similar cytotoxic effect on cells; the viability of MSCs after 3 days was 86.6 ± 3.0% compared to the control (Figure 3a). An increase in the expression of the target *BMP2* gene by 63.9 ± 3.8 times (Figure 3b) and its protein production by 5.3 ± 0.6 times compared with the control was observed 7 days after the transduction of MSC Ad-BMP2 (Figure 3c).

### 3.2. Evaluation of Osteogenic Differentiation of MSCs upon Transduction by Ad-BMP2 and upon Co-Cultivation with MSC(Ad-BMP2)

After 14 days, a significant increase in the gene expression of the osteogenic markers *Runx2*, *Spp1*, and *Bglap* compared to the control was observed in MSCs transduced with Ad-BMP2, by 6.5 ± 1.1, 6.6 ± 0.8, and 9.1 ± 3.3 times, respectively (Figure 4a–c). In MSCs co-cultured with MSC(Ad-BMP2), the increase was 11.5 ± 4.0, 2.2 ± 0.2, and 2.5 ± 0.1 times, respectively (Figure 4a–c). Additionally, there was a 2.0 ± 0.1-times increase in alkaline phosphatase activity in MSCs transduced with Ad-BMP2 and a 1.4 ± 0.1-fold increase in MSCs co-cultured with MSC(Ad-BMP2) (Figure 4d). The extracellular matrix (ECM) mineralization of MSCs was also observed (Figure 4e).

### 3.3. Assessment of Biocompatibility of Matrices In Vitro

Based on the MTT test results, it was found that PLA/PRP matrices and their components did not exhibit any cytotoxic effects on MSCs after 1 and 7 days of incubation. Moreover, when using PRP, the number of viable cells increases, and after 7 days of incubation with PRP, it was observed to be 153.5 ± 13.1%, and with PLA/PRP, it was 139.6 ± 5.4% (Figure 5a). The cells, after incubation in the presence of matrices, were alive and stained with calcein AM. There were almost no dead cells stained with DAPI (Figure 5b). Additionally, according to SEM, MSCs adhere tightly to the surface of PLA granules, spread out, and the cell density increases depending on the incubation period. The cells have a polygonal or fibroblast-like shape, which is characteristic of MSCs (Figure 5c).

### 3.4. Transducing Ability of Adenovirus Constructs Impregnated into PLA/PRP Matrices

Based on the visual GFP protein fluorescence, it was observed that Ad-GFP adenoviral constructs impregnated into PLA/PRP matrices retain their transducing ability for 14 days (Figure 6a). Ad-BMP2 was also found to be released from PLA/PRP matrices and effectively transduce MSCs. This was evidenced by an increase in target gene expression and BMP-2 protein production over 21 days (Figure 6b,c). Furthermore, the maximum expression of the *BMP2* gene was observed on day 14, which was 17.4 ± 0.4 times higher than in the control culture incubated with PLA/PRP matrices; by day 21, this indicator had decreased.

### 3.5. Osteoinductive Properties of PLA/PRP-Ad-BMP2 and PLA/PRP-MSC(Ad-BMP2) Matrices In Vitro

After 14 days, MSCs cultured with PLA/PRP-Ad-BMP2 showed a 3.2 ± 0.3-fold increase in *Spp1* gene expression, while cells in the presence of PLA/PRP-MSC(Ad-BMP2) matrices showed an 11.5 ± 2.1-fold increase, compared to control cells cultured with non-activated PLA/PRP matrices (Figure 7a). OPN protein production also increased, to 2.8 ± 0.6 ng/mL for PLA/PRP-Ad-BMP2 and to 7.8 ± 1.5 ng/mL for PLA/PRP-MSC(Ad-BMP2) (Figure 7b). Alkaline phosphatase activity increased by 1.7 ± 0.1-fold for PLA/PRP-Ad-BMP2 and by 2.2 ± 0.3-fold for PLA/PRP-MSC(Ad-BMP2) compared to PLA/PRP matrices (Figure 7c), and ECM mineralization of MSCs was also observed (Figure 7d). Therefore, incubation of MSCs with the studied matrices promotes osteogenic differentiation, with PLA/PRP-MSC(Ad-BMP2) matrices showing more pronounced osteoinductive properties compared to PLA/PRP-Ad-BMP2 matrices.

### 3.6. Osteoinductive Properties of PLA/PRP-Ad-BMP2 and PLA/PRP-MSC(Ad-BMP2) Matrices In Vivo

Using micro-CT, it was shown that implantation of all the studied matrices into the critical-size calvarial defect resulted in the formation of new bone tissue mainly from the maternal bone side; however, small islands of newly formed bone were also observed in the center of the defect. At the same time, a significant difference was found in the stimulation of neoosteogenesis; in an empty defect, the volume fraction of newly formed bone tissue (Nb.V.%) was 2.6 ± 1.5%, with implantation of non-activated PLA/PRP matrices, it was 4.5 ± 1.9%, with the implantation of control matrices PLA/PRP-Ad-GFP and PLA/PRP-MSC(Ad-GFP), it was 7.0 ± 2.4% and 15.1 ± 3.2%, respectively, and with implantation of matrices PLA/PRP-Ad-BMP2 and PLA/PRP-MSC(Ad-BMP2), it was 27.9 ± 3.5% and 33.1 ± 1.9%, respectively (Figure 8).

According to the results of histological analysis after 56 days, there were no signs of inflammation in any of the samples. The fibrin hydrogel based on PRP was replaced by collagen fibers of the connective tissue and PLA granules were surrounded by giant cells of foreign bodies (Figure 9a). It was found that the number of giant cells of foreign bodies per area of PLA granules, calculated in the fields of view, was lower in the groups with the implantation of matrices PLA/PRP-Ad-BMP2 (307 ± 58 cells/mm^2^) and PLA/PRP-MSC(Ad- BMP2) (270 ± 70 cells/mm^2^) compared to the implantation of non-activated PLA/PRP matrices (567 ± 198 cells/mm^2^). In all other groups, there were no differences in the number of giant cells of foreign bodies. According to the results of histomorphometric analysis, it was shown that the newly formed bone area (Nb.Ar) inside the defect was 2.3 ± 0.5 times higher when PLA/PRP-MSC(Ad-GFP) matrices were implanted compared to non-activated PLA/PRP matrices. There were no differences in Nb.Ar for PLA/PRP-Ad-GFP compared to PLA/PRP. At the same time, for PLA/PRP-Ad-BMP2 and PLA/PRP-MSC(Ad-BMP2), Nb.Ar was 5.6 ± 1.5 times and 10.4 ± 2.2 times higher than in the PLA/PRP group, respectively (Figure 9c).

## 4. Discussion

Despite a significant amount of data on the effectiveness of using gene-activated materials (GAMs) [2], it is possible to make a correct comparison between different technologies only under the conditions of one experiment, where only one parameter under study is changing (the composition of the material, the method of its preparation, method of impregnation of the active component, etc.), while other parameters should be unchanged.

In this work, GAMs containing fibrin hydrogel derived from PRP, adenoviral constructs with the *BMP2* gene (Ad-BMP2), or MSCs previously transduced with Ad-BMP2 were developed and compared. The material also included highly porous biocompatible PLA particles, which we previously developed for the delivery of hrBMP-2 and plasmid vectors with the *BMP2* gene, capable of providing a prolonged and gradual release of components at high therapeutic concentrations [11,19]. In this study, PLA granules served as a depot for adenoviral constructs with the *BMP2* gene and matrices for attachment of cells transduced with Ad-BMP2.

The hydrogel obtained from PRP promotes the formation of a dense clot that holds PLA granules inside it [11]. It was shown that a fibrin clot with the inclusion of PLA granules does not have a cytotoxic effect on MSCs, maintains cell adhesion, and also promotes cell proliferation (Figure 5). The ability of the PRP-based gel to initiate cell proliferation, including dose-dependent, was previously shown in a number of studies [21,22]. The presence of growth factors released during the degranulation of platelets contained in PRP affects not only cell proliferation, but also bone regeneration in general. Although PRP does not have a direct osteoinductive effect (Figure 8), it promotes the healing of bone defects by increasing bone density [18].

It was found that the most optimal conditions for adenoviral transduction are the incubation of MSCs with adenoviral vectors at 320 TCID50/mL for 24 h (Figure 1). Such conditions provide transduction of about 85% of viable cells with Ad-GFP after 3 days and lead to an increase in *BMP2* gene expression and BMP-2 protein production within a week after infection with Ad-BMP2. However, 3 days after MSC transduction under the selected conditions with viral vectors with the *GFP* gene and with the *BMP2* gene, a decrease in the number of viable cells by 15% is observed (Figure 2c and Figure 3a), which indicates the slight toxicity of the adenovirus vectors themselves that accumulate inside the cells, but not their products.

It was shown that the Ad-BMP2 and MSCs transduced with Ad-BMP2 are capable of initiating osteogenic differentiation of adipose tissue-derived MSCs in vitro (Figure 4). At the same time, on the 14th day, the expression of most osteogenic markers, the activity of Alpl and the extracellular matrix mineralization were higher when MSCs were incubated with adenovirus particles, and lower when MSCs were co-cultivated with transduced cells. This can be explained by the binding of BMP-2, synthesized by transduced cells (in the upper level of the transwell), to the receptors of the same cells, resulting in a decrease in the concentration of the protein that acts on non-transduced cells (in the lower level of the transwell).

It should be noted that the concentration of BMP-2 after the adenoviral transduction of MSCs is about 0.25 ng/mL (Figure 3c), which is several orders of magnitude lower than the concentration of rhBMP-2 used to stimulate osteogenesis. These differences are associated with the short-range action of BMP-2 and the high rate of its degradation. Thus, during gene therapy, the local concentration of BMP-2 is low, which is preferable for the patient, since high concentrations of BMP-2 can lead to soft tissue edema and other adverse effects [23].

To obtain virus-containing GAMs, adenoviral constructs or transduced cells were first impregnated into porous polylactide particles and then embedded in a fibrin gel based on autologous PRP. Previously, materials containing 3D-printed discs made of nano-calcium sulfate and alginate, fibrin gel based on PRP and MSCs modified with adenoviral constructs with the *BMP2* gene were developed [21]. In our study, the use of highly porous PLA granules capable of adhering a large number of cells to the walls of pores made it possible to impregnate 10 million cells per matrix, which is an order of magnitude higher than in the previous experiment. Furthermore, using porous PLA particles, which have a higher elastic modulus than the PRP hydrogel, will promote osteogenic differentiation of MSCs [24].

It has been shown that adenoviral particles are able to be released from the matrices and infect MSCs, as evidenced by the presence of the fluorescent GFP protein in cells after transduction with Ad-GFP and an increase in the expression level of the *BMP2* gene and concentration of its product after infection with Ad-BMP2 (Figure 6).

A study of the osteogenic properties of GAMs with Ad-BMP2 and MSCs transduced with Ad-BMP2 showed that all materials have an osteoinductive effect both in vitro and in vivo. Incubation of MSCs in the presence of PLA/PRP-Ad-BMP2 or PLA/PRP-MSC(Ad-BMP2) results in an increase in *Spp1* gene expression and osteopontin protein production, Alpl activity, and ECM mineralization. However, matrices containing transduced cells cause a more significant increase in these parameters. When the studied matrices PLA/PRP-Ad-BMP2 or PLA/PRP-MSC(Ad-BMP2) were implanted into critical-size calvarial defects of rats, the formation of young bone tissue was observed after 56 days, which formed from the edges of the defect towards its center. According to micro-CT data, the efficiency of neoosteogenesis was 28% with PLA/PRP-Ad-BMP2 implantation and 33% with PLA/PRP-MSC(Ad-BMP2) implantation (Figure 8). The newly formed bone area upon implantation of these matrices is significantly higher than when using GAMs with the *GFP* gene. At the same time, the number of giant cells of foreign bodies surrounding PLA granules, on the contrary, is lower when matrices with the *BMP2* gene were implanted (Figure 9), which may indicate the immunomodulatory effect of BMP-2 on macrophages, which enhances regeneration [25].

Thus, in the course of a comparative study, it was found that PLA/PRP-MSC(Ad-BMP2) have more pronounced osteoinductive properties compared to PLA/PRP-Ad-BMP2 matrices. This can be explained by the fact that during ex vivo gene therapy, the target protein will begin to be released from the transduced cells inside the materials almost immediately after transplantation, and its concentration will quickly reach therapeutic values. And during in vivo gene therapy, not all adenoviral constructs released from matrices will be able to enter target cells, and secretion will be somewhat delayed, which will reduce the concentration of the target protein.

Earlier, in a similar comparative study, it was shown in one experiment that transplantation of MSCs transduced with Ad-BMP2 promotes the healing of a rat skull defect more effectively than the implantation of cells incubated in an osteogenic medium [26]. In addition, using MSCs transduced with Ad-BMP2 was shown to be convincingly superior to cell transplantation with recombinant BMP-2 in a rat femoral defect model [27].

Furthermore, adenoviral therapy for bone regeneration has proven to be more preferable than other genetic methods. For example, Blum et al. demonstrated that MSCs genetically modified with Ad-BMP2 have a higher osteogenic capacity then cells infected with lentiviral constructs or lipoplexes [28]. At the same time, adeno-associated viruses have significant limitations for gene therapy of bone regeneration due to their low ability to transduce osteogenic precursors such as bone marrow-derived MSCs [29] and the high variability in infection efficiency from donor-to-donor [30].

However, the effectiveness of adenovirus-based therapy may be reduced if the patient has pre-existing antibodies against the virus [31]. To solve this problem, studies are underway that are aimed at isolating transplanted cells with adenoviral vectors inside scaffolds that are impermeable to immune cells [32,33]. Additionally, methods of genetic modification of viral vectors and approaches aimed at a local decrease in the immune response can be used [6]. In this regard, ex vivo gene therapy has advantages over the direct administration of adenoviruses, as it involves the use of MSCs, which are a source of anti-inflammatory components that can temporarily reduce the immune response and increase the efficiency of regeneration.

## 5. Conclusions

During the study, GAMs were developed based on adenoviral constructs with the *BMP2* gene or MSCs transduced with the *BMP2* gene, highly porous polylactide particles, and a fibrin clot obtained from PRP. Both types of matrices have osteogenic potential in vitro and in vivo. However, ex vivo delivery of the target gene demonstrated greater efficiency compared to in vivo delivery, which makes it a more promising approach for clinical use in the treatment of bone defects.

## Figures and Tables

**Figure 1 cells-12-01762-f001:**
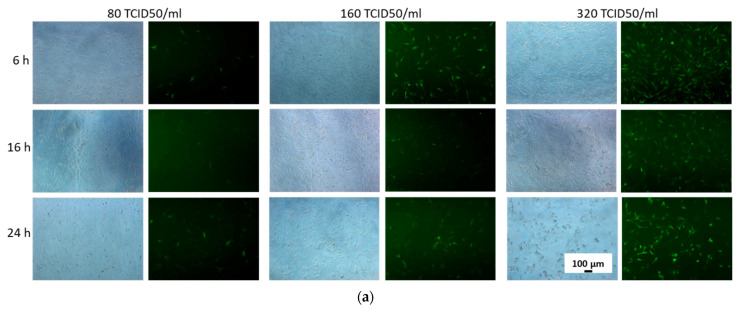
Efficiency of adenoviral transduction of MSCs depending on incubation time and viral load after 1 (**a**,**c**) and 3 days (**b**,**d**): phase contrast (left column) and fluorescence (right column) microscopy; (**c**,**d**) percentage of transduced cells, flow cytometry.

**Figure 2 cells-12-01762-f002:**
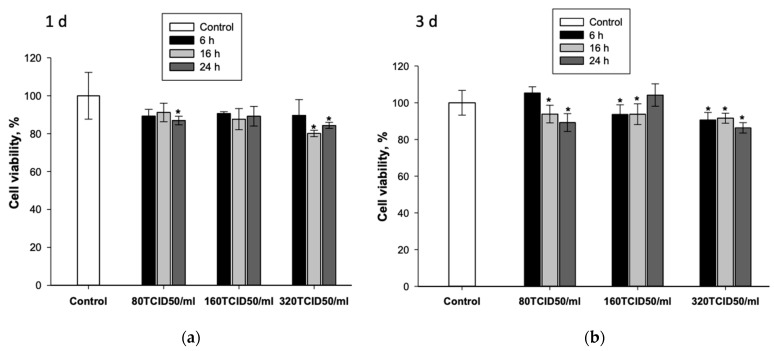
Viability of MSCs 1 day (**a**) and 3 days (**b**) after adenoviral transduction depending on incubation time and viral load, MTT test. * *p* < 0.05 (relative to control).

**Figure 3 cells-12-01762-f003:**
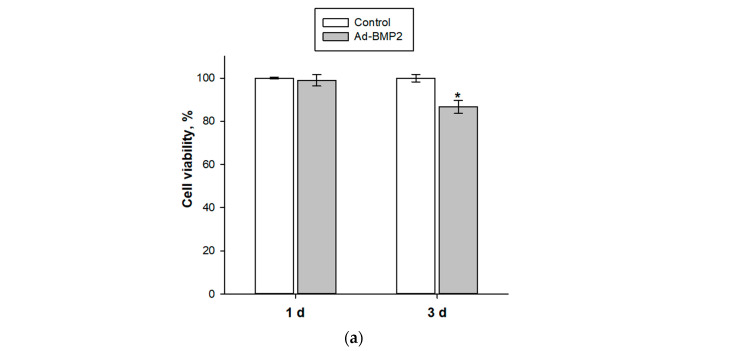
Efficiency of MSC transduction by Ad-BMP2: (**a**) viability of MSCs after adenoviral transduction, MTT test; (**b**) relative *BMP2* gene expression, RT-PCR; and (**c**) BMP-2 protein production, ELISA. * *p* < 0.05 (relative to control).

**Figure 4 cells-12-01762-f004:**
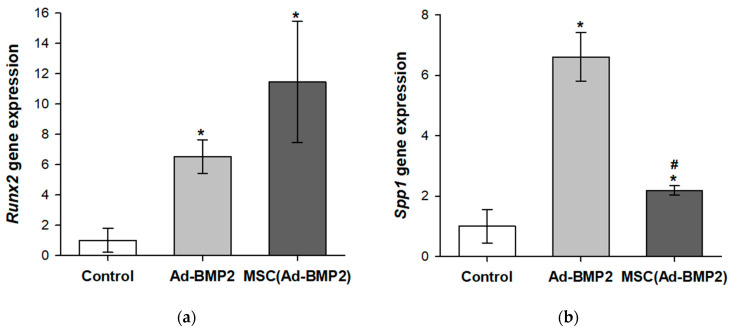
Induction of osteogenic differentiation of MSCs upon Ad-BMP2 transduction and upon co-cultivation with transduced MSC(Ad-BMP2) after 14 days: (**a**) relative *Runx2* gene expression, RT-PCR; (**b**) relative *Spp1* gene expression, RT-PCR; (**c**) relative *Bglap* gene expression, RT-PCR; (**d**) Alpl activity, spectrophotometry. * *p* < 0.05 (relative to control), # *p* < 0.05 (relative to Ad-BMP2); and (**e**) ECM mineralization of MSCs, alizarin red staining, light microscopy.

**Figure 5 cells-12-01762-f005:**
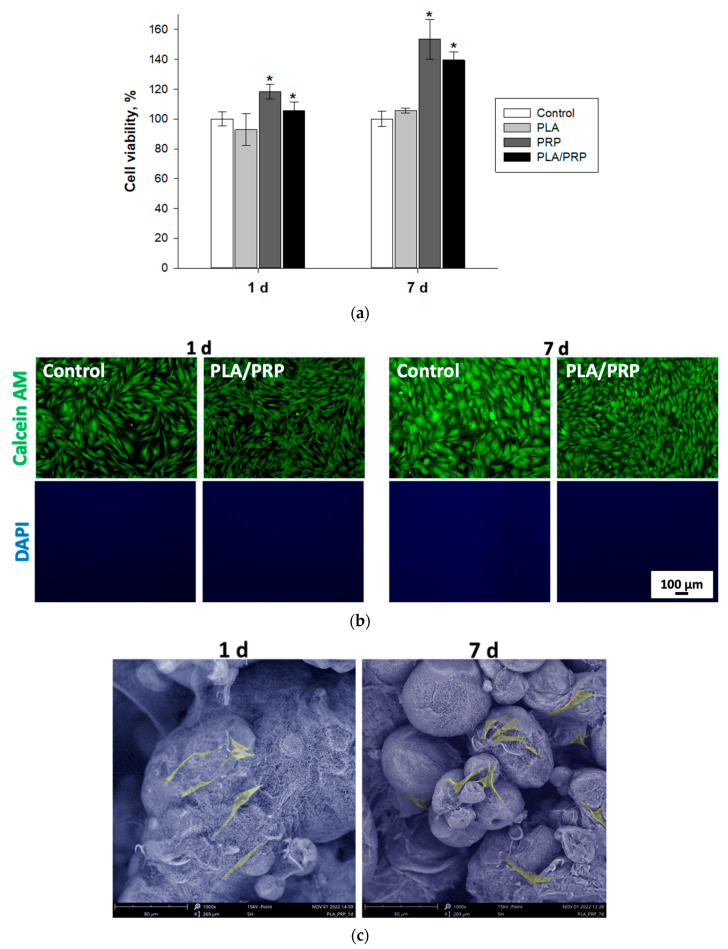
Cytocompatibility of PLA/PRP matrices in vitro: (**a**) viability of MSCs 1 and 7 days after incubation with PLA/PRP matrices and its components, MTT test. * *p* < 0.05 (relative to control); (**b**) assessment of cytotoxicity of PLA/PRP matrices using fluorescence microscopy; and (**c**) adhesion of MSCs to PLA/PRP matrices, SEM, 1000×.

**Figure 6 cells-12-01762-f006:**
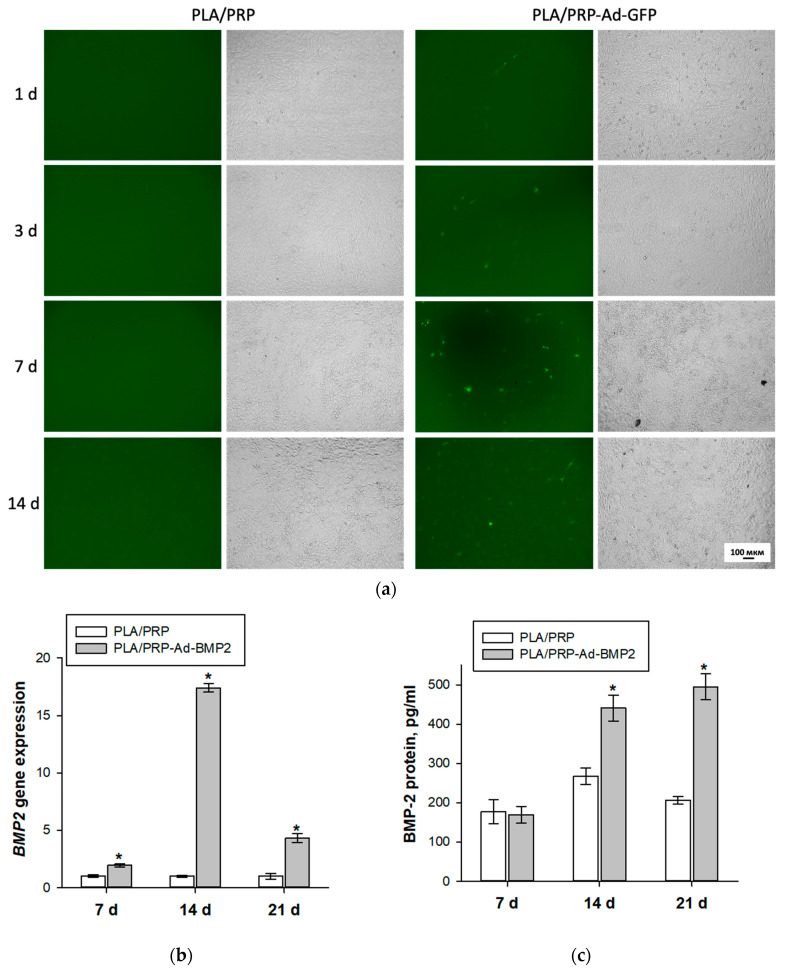
Transducing ability of adenovirus constructs impregnated into PLA/PRP matrices: (**a**) transduced MSCs, after incubation with PLA/PRP-Ad-GFP matrices (green), fluorescence (left column) and phase contrast (right column) microscopy; (**b**) relative *BMP2* gene expression after incubation of MCSs with PLA/PRP-Ad-BMP2, RT-PCR; and (**c**) BMP-2 protein production after incubation of MCSs with PLA/PRP-Ad-BMP2, ELISA. * *p* < 0.05 (relative to PLA/PRP matrices).

**Figure 7 cells-12-01762-f007:**
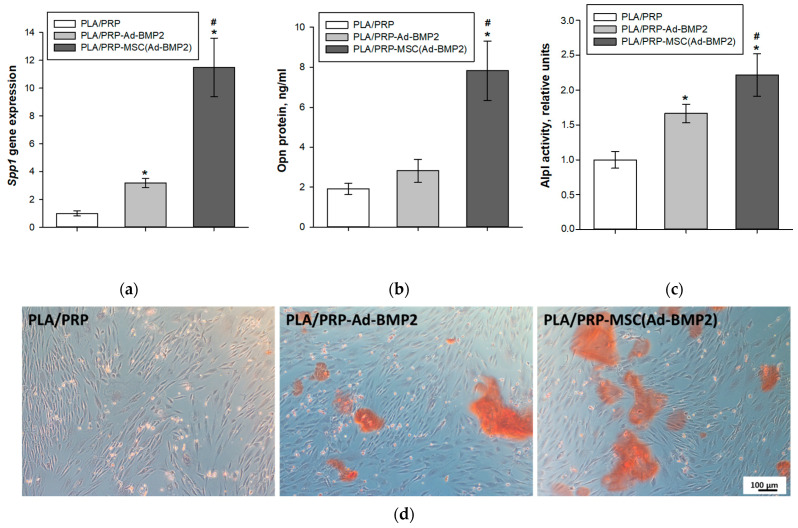
Osteogenic differentiation of MSCs 14 days after incubation with PLA/PRP-Ad-BMP2 и PLA/PRP-MSC(Ad-BMP2) matrices: (**a**) relative *Spp1* gene expression, RT-PCR; (**b**) Opn protein production, ELISA; (**c**) Alpl activity, spectrophotometry. * *p* < 0.05 (relative to PLA/PRP matrices), # *p* < 0.05 (relative to PLA/PRP-Ad-BMP2); and (**d**) ECM mineralization of MSCs, alizarin red staining, light microscopy.

**Figure 8 cells-12-01762-f008:**
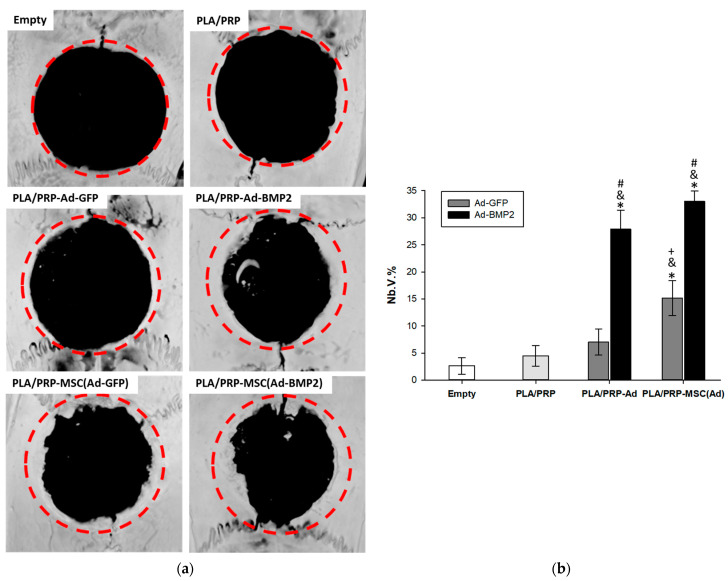
Critical-size bone defect regeneration in rats: (**a**) 3D reconstructive images at 8 weeks after matrix implantation, micro-CT; (**b**) the volume of newly formed bone (Nb.V.%) measured using micro-CT. * *p* < 0.05 (relative to empty defect) and *p* < 0.05 (relative to PLA/PRP matrices), # *p* < 0.05 (relative to PLA/PRP matrices impregnated with Ad-GFP or MSC(Ad-GFP), respectively), + *p* < 0.05 (relative to PLA/PRP matrices impregnated with adenoviruses without cells).

**Figure 9 cells-12-01762-f009:**
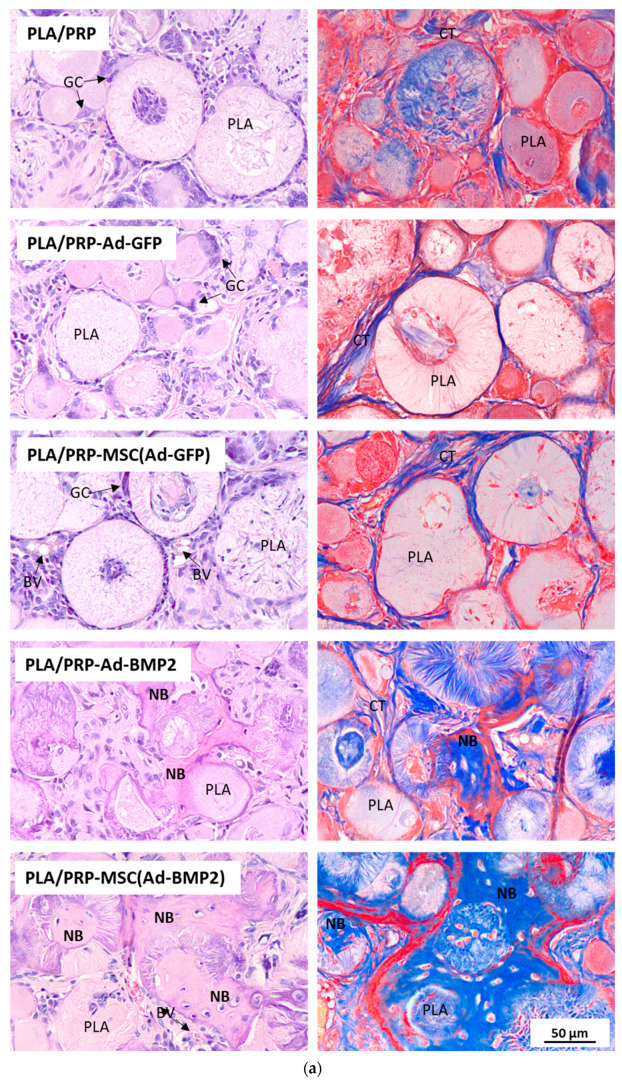
Bone regeneration in vivo: (**a**) histologic sections at 8 weeks after matrix implantation, H&E (left column) and Masson’s trichrome staining (right column). Light microscopy. BV—blood vessels, CT—connective tissue, GC—foreign body giant cells, NB—newly formed bone; (**b**) quantitative assessment of the resorption degree of the matrices; and (**c**) the newly formed bone area (Nb.Ar) inside the defect measured using histomorphometrical analysis. * *p* < 0.05 (relative to PLA/PRP matrices), # *p* < 0.05 (relative to PLA/PRP matrices impregnated with Ad-GFP or MSC(Ad-GFP), respectively), + *p* < 0.05 (relative to PLA/PRP matrices impregnated with adenoviruses without cells).

## Data Availability

Available upon request.

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
