# Peer review of "Adenovirus-Based Gene Therapy for Bone Regeneration: A Comparative Analysis of In Vivo and Ex Vivo BMP2 Gene Delivery"

_cells, 2023, doi:10.3390/cells12131762_

Round 1
Reviewer 1 Report
This work refers to adenovirus-mediated gene therapy in bone regenerative medicine. It is overall well written, however some adjustments are required:
- Introduce the numbers in the subsections of the methods
- In the section 3.1 add the letters in the text to describe the figure (ie Fig.1a)
-The authors have been described only the figs 1C and 1d. I reccomend to describe also the fig.1a and 1b
-In the figure 2, add the letters A and B. Then, re-use them in the text
-I suggest quantifying the Alizarin red staining in order to obtain the quantification of calcium deposits. For example, the authors may use the CPC
-In the fig.5c the magnification is only in the image. I suggest adding it in the caption
-Add reference in lines 449-454
-Add a paragraph for the conclusion
The English language is good. Few errors are in the text. A minor editing is required.
Reviewer 2 Report
This is well-written manuscript that compares the efficacy of treating a rat calvarium bone defect with BMP2 expressing adenovirus vs MSCs transduced with the same BMP2 expressing adenovirus. The in vivo defect model also employs clotted PRP and PLA granules. The roles of the PRP and the PLA granules in the experiments is not controlled for as emphasis was placed on in vivo vs ex vivo Ad-BMP2 effects. Consistent with several prior studies, ex vivo transduction of MSCs with Ad-BMP2 appeared to promote more bone formation in vivo than when Ad-BMP2 was used as a gene therapy in vivo as per histomorphometry though the uCT analysis failed to show any significant effect..
The direct comparison of in vivo to ex vivo Ad-BMP2 therapy does provide some novel findings. The manuscript could be improved by focusing on those differences and considering some additional interpretations of the results. For instance, the uCT data indicates that Ad-BMP2 in vivo (PLA/PRP-Ad-BMP2) produced 4X more bone than the control (PLA/PRP Ad-GFP) while the ex vivo Ad-BMP2 transduced MSCs only had an approximate 2 fold effect (PLA/PRP-MSC(Ad-BMP2) vs PLA/PRP-MSC(Ad-GFP); Figure 8). Is this improved performance of the PLA/PRP-Ad-BMP2 because of the PLA particles, the PRP, or both?
Other improvements:
1. The matrices used were contained multiple components that were not adequately controlled. Specifically, whether the PLA particles aid in Ad-BMP2 transduction, retention at the implantation site or other property? What effects of the PRP have on Ad-BMP2 transduction/activity or osteogenesis in vivo? Failing these experiments, the investigators should appropriately qualify their descriptions of the experimental results.
2. Reduce the number of figures describing the Ad-BMP2 transduction efficiencies and post infection viability and focus more on BMP2 production.
3. Discussion should include a comparison of how much BMP2 is being made by Ad-BMP2 vs rhBMP-2 doses used to promote osteogenesis in MSC cultures and in vivo.
4. Sentence starting on line 59 should have a reference.
5. Discussion should provide an explanation as to why no islands of ectopic bone formed within the calvaria defect, particularly with the MSC-Ad-BMP2 treated defects. If BMP2 is osteo-inductive as stated in sentence beginning on line 447, then islands of bone should have formed in the defect and not just from around the edges.
6. The histomorphometry was performed using 20 fields of view for each specimen. However, a 7mm diameter defect is a very large area and based on the uCT images, it would appear that where the 20 fields of view were made would significantly impact the results. Fields of view close to or at the edges would have bone, fields of view in the middle of the defect would have no bone. A more systemic approach to the histomorphometric analysis should be undertaken to confirm (or refute) the uCT image analysis.
No comments, fairly well-written.
Round 2
Reviewer 2 Report
Prior research has shown that in vitro transduction of MSCs with BMP2 expression systems, generally produces a better result in vivo. Here, the in vivo response between Ad-BMP2 vs MSC-AdBMP2 were statistical not different and the ration of bone formed relative to matching control indicated that PLA/PRP mix may be promoting better in vivo Ad-BMP2 transduction. This was the point brought up in the review, but apparently was not clear.
Aside from the point above, the authors have satisfactorily addressed the prior comments.